# An Overview of the Most Commonly Used Methods for the Detection of *Nosema* spp. in Honeybees

**DOI:** 10.3390/microorganisms13112501

**Published:** 2025-10-31

**Authors:** Imrich Szabó, Monika Sučik, Jana Morochovičová, Lucia Sabová

**Affiliations:** 1Department of Biology and Physiology, University of Veterinary Medicine and Pharmacy in Košice, 04181 Košice, Slovakia; jana.morochovicova@student.uvlf.sk; 2Department of Pharmatology and Toxicology, University of Veterinary Medicine and Pharmacy in Košice, 04181 Košice, Slovakia; lucia.sabova@uvlf.sk

**Keywords:** honeybee, *Nosema apis*, *Nosema ceranae*, molecular analysis, PCR

## Abstract

Nosemosis is a disease caused by microsporidia, which are strictly intracellular pathogens, currently considered to be most closely related to fungi. These microscopic parasites infect a variety of hosts, significantly affecting honeybees (*Apis mellifera*). Nosemosis is one of the most serious diseases of bees and is caused primarily by two species: *Nosema apis* and *Nosema ceranae*. This infection adversely affects the digestive tract of the bees, causes a reduction in their vitality, and can lead to the death of entire colonies. The diagnosis of nosemosis has undergone extensive development. Traditionally, the identification of microsporidia was performed by examination of bee digestive tract (macerated) by light microscopy. Transmission electron microscopy (TEM) and scanning electron microscopy (SEM) are expensive methods that require skilled personnel and were used only when high resolution was necessary. Modern methods, such as polymerase chain reaction (PCR), allow detection of infection at species and genotype levels, thereby increasing the accuracy of diagnosis. Despite advances in molecular techniques, research into nosemosis still faces challenges. This review focuses on a comparison of different diagnostic techniques and their pitfalls that can be integrated into strategies to combat nosemosis and protect the health of honeybee colonies.

## 1. Introduction

The honeybee (*Apis mellifera*) is an inseparable part of nature. They live in colonies (consisting of several tens of thousands of individuals) that function as superorganisms. Their most important role is pollination of flowering plants, including fruit trees and crops, and, therefore, they play a major and indispensable role in agriculture as well as in the conservation of ecological biodiversity in general [1].

Extensive research has shown that multiple stressors, such as forage loss, pathogens, parasites, agropesticides, and improper beekeeping practices, are involved in bee population declines [2,3,4].

Globally, microsporidian species *Nosema apis* and *N. ceranae* are currently the most prevalent pathogenic microorganisms of *A. mellifera* [5,6,7,8,9,10,11]. The spores of these two species are very similar at first sight; the differences between them can be distinguished only under transmission electron microscopy—TEM—in dimensions and number of polar filament coils [5] or by surface structure under scanning electron microscopy—SEM [12]. However, recent studies [13,14,15,16] show that *N. ceranae* infection on honeybee colonies is more aggressive compared to *N. apis* infection, as it is a relatively new pathogen for the honeybee. *N. neumanni*, genetically close to *N. apis*, was recently described as infecting European honeybees in Uganda, but information on this *Nosema* species is scarce [17].

This review provides a comparative analysis of various diagnostic methods used for detecting nosemosis, emphasizing their respective advantages and limitations. Particular attention is given to the potential challenges and inaccuracies that may arise when applying these techniques. It is also underlined that the integration of diagnostic techniques plays a key role in improving strategies to control nosemosis and to safeguard the vitality and survival of honeybee colonies.

## 2. Overview of Methodologies Used for the Detection of *Nosema* spp.

Reliable and harmonized diagnostics are crucial to ensure the quality of follow-up and research results. For this reason, the first European interlaboratory comparison (ILC) was organized in 2017 to assess both the methods and the results obtained by the national reference laboratories (NRLs) for spore counts of *Nosema* spp. using the microscope published by Duquesne et al. (2021) [18]. The assessment included specificity, sensitivity, trueness, and accuracy. Quantitative results were analyzed in accordance with the international standards NF ISO 13528 (2015) [19] and NF ISO 5725-2 (1994) [20]. The overall results indicate a global specificity of 98% and a global sensitivity of 100%, demonstrating the advanced performance of microscopic methods applied to *Nosema* NRL spores. The study therefore concluded that the use of microscopy for the detection and quantification of *Nosema* spp. spores is sufficiently reliable.

However, light microscopy examination is mainly used as a basic preliminary method to check for the presence of *Nosema* spp. spores [21]. Previous studies have highlighted the efficacy of advanced microscopy methods, such as transmission electron microscopy (TEM) and scanning electron microscopy (SEM), in providing detailed morphological views that surpass traditional light microscopy [5,12].

Nevertheless, to identify *Nosema* species more accurately, it is essential to use PCR-based molecular methods, as the differences between *N. ceranae* and *N. apis* spores are difficult to detect and can often complicate diagnosis, especially in mixed infections. In particular, multiplex PCR, PCR-RFLP, and qPCR or real time are used in research, which allow accurate detection and analysis of genetic material in a variety of biological and clinical applications.

The first application of multiplex PCR for the detection of *Nosema* spp. in honeybees (*Apis mellifera*) was published by Martín-Hernández et al. in 2007 [22]. They developed a method to simultaneously identify *N. apis* and *N. ceranae* by amplifying specific regions of the 16S rRNA gene. This technique allowed accurate identification of both species in purified spores and in bee homogenates, which contributed to a better understanding of the distribution of *N. ceranae* in Europe.

In 2010, Hamiduzzaman et al. [23] introduced a semi-quantitative triplex PCR method for the diagnosis and quantification of *Nosema* infections in bees. This method allowed the detection of low levels of infection, up to 100 spores per bee, and provided a sensitive tool for laboratories without access to real-time PCR technologies. In addition to the detection of *N. apis* and *N. ceranae*, this method included amplification of the ribosomal protein gene RpS5 from the bee as an internal control.

These studies laid the foundation for the use of multiplex PCR in the diagnosis of Nosema infections in bees, allowing more efficient and accurate detection of these pathogens.

The Table 1, Table 2 and Table 3 below contain various primers and their sequences used in molecular studies to identify and characterize *Nosema* species using multiplex (including duplex) PCR, single species-specific PCR, and RFLP PCR methods.

Table 1, Table 2 and Table 3 show the wide range of primers used to detect different *Nosema* species and to analyze their genetic material. The primers are designed to amplify specific DNA sequences that allow identification of pathogens at the level of molecular biology. Each primer is accompanied by a reference to the study that used or developed the primer.

Early authors of studies on real-time PCR for the detection of *Nosema* spp. in honeybees (*Apis mellifera*) include Bourgeois et al. (2010) [30] and Forsgren and Fries (2010) [31]. They developed a method for the genetic detection and quantification of *N. apis* and *N. ceranae* using real-time PCR. Both studies have contributed to significant advances in the molecular diagnosis of these parasites.

Subsequently, in 2019, Li et al. [32] presented an improved method combining rapid and simple DNA extraction with quantitative real-time PCR for high-throughput molecular screening of *Nosema* spp. in Antheraea pernyi. This method allows rapid and automated DNA preparation and is cost-effective.

In 2021, Truong et al. [33] developed ultra-rapid quantitative PCR (UR-qPCR) for the rapid detection of *N. ceranae* in honeybees, allowing quantification of infection in approximately 20 min.

Ptaszyńska et al. (2014) [12] described the development and validation of an LAMP assay for the detection and differentiation of two origins of nosemosis in honeybees, *N. apis* and *N. ceranae*. The method was based on the use of six special primers recognizing sequences in 16S rDNA, with the reaction taking place at a constant temperature of 60 °C and the results could be obtained after only 30 min. LAMP requires following a set of four to six primers: forward inner primer (FIP), backward inner primer (BIP), forward outer primer (F3), backward outer primer (B3), and additionally forward loop primer (LF) and backward loop primer (LB), which significantly reduces the time of the amplification. LAMP detected *N. apis* and *N. ceranae* target DNA down to the total DNA concentration of 100 fg, whereas the standard PCR detected these DNA samples to the total DNA concentration of 100 pg. Therefore, LAMP appeared to be 10^3^-fold more sensitive than standard PCR in *N. apis* and *N. ceranae* detection. All honeybee DNA samples identified by loop-mediated isothermal amplification reaction as containing *N. apis* and *N. ceranae* DNA or these microsporidia’s DNA were positively supported by standard PCR (321 bp and 218–219 bp for *N. apis* and *N. ceranae*, Martín-Hernández 2007) [22]. It should be emphasized that the LAMP described by Ptaszyńska et al. (2014) [12] allows not only identifying but also distinguishing between *N. apis* and *N. ceranae* infections in honeybees.

These advances in real-time PCR techniques have contributed significantly to the accurate and rapid detection and quantification of *Nosema* spp. in bees and other hosts.

Table 4 provides an overview of the primers used with a specific sequence (5′ → 3′), with different fragment sizes targeting different target genes, using real-time PCR to detect different *Nosema* species, namely *N. apis*, *N. ceranae*, and *N. bombi*.

Table 5 shows the different PCR conditions used in studies aimed at detecting *Nosema* species. The main difference between studies is the annealing temperature, which varies by author. For instance, while Klee et al. (2007) [29] used a temperature of 48 °C and Kunat-Budzynska, Łabuc, and Ptaszynska (2025) [37] 46 °C, Webster et al. (2004) [27] and Sgroi, D’Auria et al. (2025) [38] set the temperature at 62 °C, while other studies (e.g., Martín-Hernández et al. (2007) [22] and Stevanovic et al. (2011) [28]) chose a temperature of 61.8 °C. These differences in temperature may be due to the use of different types of polymerases or PCR mixes, as well as different primers or sample types used for amplification. Another parameter that varies is the annealing time—most authors, such as Fries et al. (2013) [24] and Ostroverkhova et al. (2020) [39], use 30 s, but studies such as Klee et al. (2007) [29] and Webster et al. (2004) [27] set a longer time of 1 min, which may indicate the need for stronger primer binding under specific conditions. The number of cycles ranges from 30 to 45 cycles, which affects the overall DNA amplification—more cycles are used in studies that target smaller amounts of DNA or more challenging samples. The volume of the PCR mix varies from 10 μL to 50 μL, which may be related to different experimental requirements for the amount of reaction mix. These variations show that different authors tailor PCR conditions according to the specific objectives of their experiments, be it sample type, primer type, or detection sensitivity requirements.

Table 6 compares the different PCR protocols used in the studies for the detection of *Nosema* spp. Each protocol differs in the length and temperature of the initial activation step and the number of PCR cycles, as well as the different denaturation, annealing, and extension steps. These differences may affect the sensitivity and specificity of detection.

## 3. Comparison of Methodologies by Our Department

Our collective is engaged in research where we use different methodologies to detect *N. apis* and *N. ceranae* species in honeybee carcass samples. The bee samples are sent by the beekeepers themselves to the Institute of Beekeeping in Liptovský Hrádok, where microscopic analysis is carried out. Subsequently, the samples are examined by molecular methods at the University of Veterinary Medicine and Pharmacy in Košice.

### Molecular Diagnostics

We isolated DNA from crushed bee abdomen samples using the commercially available DNA -sorb- AM isolation kit from AmpliSense and following the manufacturer’s instructions. We used a PRECELLYS tissue homogenizer with a program of 6500 rpm for 2 × 45 s to break the hard shell of the spores.

In the next step, we created a PCR mix for each sample, which consisted of PCR water, master mix, and 10 µM mediums for both species of *Nosema* spp. (Table 7). We added the template to the mix and vortexed thoroughly.

Duplex PCR was performed using a VWR RISTRETTO thermal cycler (VWR International, Radnor, PA, USA). The initial denaturation took place for 4 min. at 95 °C, followed by 28 cycles of denaturation at 95 °C (25 s), annealing at 58 °C (45 s), and polymerization at 72 °C (2 min). The final extension step lasted 7 min. at 72 °C. To prove the presence of DNA in the investigated samples, we used gel electrophoresis, using a 1.5% agarose gel. After evaluating the results with a UV transilluminator, the DNA concentration was measured with a NanoDrop and the PCR products were sent for sequencing. The obtained sequences were compared with the sequences stored in the gene bank using the BLAST program.

For the detection of *Nosema* spp. from bee samples using real-time PCR, we used a PCR mix containing SYBR Green. The total volume of the reaction mixture was 25 μL (Table 8), containing the components listed in Table 8.

PCR reactions were performed according to (Malčeková et al., 2013) [41] in a thermal cycler (LightCycler 480 II, Roche, Basel, Switzerland). The program consisted of the following steps: incubation, initial denaturation, 40 cycles of denaturation and hybridization, and melting (melting temperatures) (Table 9).

In Table 10, we report the yield of each methodology, i.e., we compare the same sample of 500 bee colonies. Most of the positive samples were detected by microscopy, which also seems to be the least expensive. Duplex PCR shows the least number of positive samples, which may be due to freezing and thawing of samples or human factor failure. Compared to real-time PCR, duplex is more time-consuming, and the funds required to provide these methods are approximately the same.

## 4. Discussion

Microscopic analysis of nosemosis in honeybees is one of the most commonly used detection methodologies. It is used both by beekeepers themselves and by research centers. It requires minimal funding and takes a short time to implement. However, microscopic detection is limited to observation and counting of spores, with visible spores representing only a fraction of the life cycle of microsporidia, which can survive outside the host gut cells [42]. Because other stages occurring outside the organ of observation may remain invisible, these apparently low levels of infection may thus be difficult to detect. In our experience, microscopy appears to be a reliable method for detecting nosemosis, but it is not sufficient for accurate species and genotype identification.

In molecular detection of parasites using PCR, several critical parameters must be considered to ensure accuracy and specificity. One of the most important is the number of amplification cycles. In our study, we used 28 cycles, which are below the threshold where false positives may increase. Several publications indicate that exceeding 30 cycles can lead to amplification of non-specific products or contaminants, resulting in misinterpretation of results. Therefore, it is essential to optimize the number of cycles based on the quantity and quality of DNA in the sample [24,43].

Another key parameter is the annealing temperature, which affects the specificity of primer binding to the target sequence. Higher annealing temperatures improve specificity but may reduce amplification efficiency if not properly adjusted. In our protocol, we used 58 °C, representing a compromise between specificity and efficiency. Differences in annealing temperatures across studies (e.g., from 46 °C to 62 °C) highlight the need for individual optimization depending on the primers and polymerases used [22,27,29,37,38].

Amplicon length and nucleotide diversity also play a crucial role. Shorter amplicons are suitable for degraded samples but may offer lower taxonomic resolution. Longer sequences provide more genetic information, which is beneficial for species differentiation. In our work, we used amplicons ranging from 218 to 321 bp, which proved effective for detecting *N. apis* and *N. ceranae* [12,22].

Also, Baigazanov et al. (2022) [44], in their research results, point out the need for further research on the use of molecular biology methods for the needs of distinguishing the species causing nosemosis infection (PCR).

Truong et al. (2021) [33] also concluded in their study that quantification of *N. ceranae* by UR-qPCR improved the sensitivity and stability of detection compared to microscopic counting. By molecular quantification, they calculated the total amount of *Nosema* DNA in the sample according to a standard curve. Thus, PCR detection using specific primers proved to be more useful for quantitative detection of intracellular parasites. PCR detection has also been chosen by McAfee et al. (2024) [45] to extensively investigate common honeybee pathogens in the context of blueberry pollination.

Since the currently widely used PCR and microscopy are expensive techniques that require skilled personnel and may not be available in every laboratory, it is appropriate to look for cheaper and less labor-intensive methodologies. Study [46] describes the production and characterization of four monoclonal antibodies (mAbs) against *N. ceranae* and *N. apis* and the development of an indirect fluorescent antibody test (IFAT). The IFAT using mAbs was compared with microscopy and PCR for 180 hive samples. The diagnostic test revealed similar percentages of sensitivity and specificity as IFAT (97.79% and 93.18%) and microscopy (97.79% and 95.45%), with 100% for PCR considered the “gold standard”. mAb (7D2) has been patented for its high specificity for *N. ceranae*. IFAT using mAb therefore appears to be a good alternative to microscopy and PCR in laboratories where PCR is not available for the detection and identification of both *Nosema* spp.

In the submitted manuscript, I have decided to use the names *Nosema ceranae* and *Nosema apis*, although some current taxonomic databases list these species under the genus *Vairimorpha*. This decision is based on both professional and practical reasons. The name *Nosema* has long been established in the field of bee disease research, as well as in apicultural practice, and its use contributes to terminological consistency and clarity of the text for the wider professional community, including veterinary workers, beekeepers, and students. The taxonomic revision, which proposed the transfer of the species *Nosema apis* and *Nosema ceranae* to the genus *Vairimorpha* [43], was based mainly on the analysis of a limited number of molecular markers. This approach has provoked extensive professional debate about its justification, as several authors point to the need for a more comprehensive phylogenetic analysis before adopting such fundamental nomenclatural changes. Bartolomé et al. (2024) [47], for example, warn that such a revision may lead to ambiguities in the interpretation of older data and unnecessary confusion in the literature on bee pathogens. Several recent publications focusing on applied research and diagnosis of nosemosis also prefer to retain the original nomenclature *Nosema* [37,38], as this designation is still commonly used in laboratory practice, diagnostics, and professional recommendations, and its retention helps to minimize terminological chaos during the transitional period of taxonomic revision.

## 5. Conclusions

The aim of this work was to provide an overview of the diagnostic capabilities of an important honeybee parasite that is crucial in the face of an alarming worldwide decline in honeybee colony numbers. Honeybees are not only of great economic importance but also provide invaluable ecological benefits, so it is essential to identify and analyze the factors that are leading to their decline so that we can implement effective measures to protect these important pollinators.

One of the important factors causing bee decline is nosemosis, which contributes to the gradual collapse of bee colonies by weakening their defenses. Infections caused by parasites of the genus *Nosema* can cause widespread problems in colonies, which, in turn, affect the overall health and productivity of the colonies.

Microscopic determination plays an important role in the diagnosis of nosemosis and is suitable for the rapid detection of *Nosema* in bee colonies, which is crucial for the early detection of the disease and the prevention of its spread. However, modern molecular methods such as polymerase chain reaction (PCR) are desirable for species differentiation. PCR is not only a very sensitive method but also highly specific, meaning that it can recognize different *Nosema* species even at low concentrations. These methods are increasingly used in the diagnosis of bee diseases and provide valuable information for research and the protection of colonies from these serious pathogens. A relatively new methodology is IFAT using mAb, which can identify specific *Nosema* species and is also more affordable than PCR.

## Figures and Tables

**Table 1 microorganisms-13-02501-t001:** Detection of *Nosema* by multiplex PCR of partial 16S rRNA gene.

Species	Type of PCR	Primers	Sequence (5′ → 3′)	Fragment (bp)	References
*N. ceranae*	duplex PCR	218MITOC-FOR	CGGCGACGATGTGATATGAAAATATTAA	218–219	[22]
*N. ceranae*		218MITOC-REV	CCCGGTCATTCTCAAACAAAAAACCG		
*N. apis*		321APIS-FOR	GGGGGCATGTCTTTGACGTACTATGTA	321	
*N. apis*		321APIS-REV	GGGGGGCGTTTAAAATGTGAAACAACTATG		
*N. apis*	multiplex PCR	Mnapis-F	GCATGTCTTTGACGTACTATG	224	[24]
*N. bombi*		Mnbombi-F	TTTATTTTATGTRYACMGCAG	171	
*N. ceranae*		Mnceranae-F	CGTTAAAGTGTAGATAAGATGTT	143	
*Uni*		Muniv-R	GACTTAGTAGCCGTCTCTC		
*N. ceranae*	multiplex PCR	Mnceranae-F	CGT TAA AGT GTA GAT AAG ATG TT	143	[25]
		Mnceranae-R	GAC TTA GTA GCC GTC TCT C		
*N. apis*		Mnapis-F	GCA TGT CTT TGA CGT ACT ATG	224	
		Mnapis-R	GAC TTA GTA GCC GTC TCT C		
*N. apis*	multiplex PCR		CCATTGCCGGATAAGA GAGT	401	[26]
			CACGCATTGCTGCATCA TTGAC		
*N. ceranae*			CGGATAAAAGAGTCC GTTACC	250	
			TGAGCAGGGTTCTAGGGAT		

**Table 2 microorganisms-13-02501-t002:** Detection of *Nosema* spp. by *N. apis*-specific PCR.

Species	Type of PCR	Primers	Sequence (5′ → 3′)	Fragment (bp)	Target	References
*N. apis*	*N. apis*-specific PCR	NosA-F	CCGACGATGTGATATGAGATG	209	16S rRNA	[27]
*N. apis*		NosA-R	CACTATTATCATCCTCAG ATCATA			

**Table 3 microorganisms-13-02501-t003:** Differentiation between *Nosema* species based on PCR-RFLPs of partial 16S rRNA gene.

Type of PCR	Primers	Sequence (5′ → 3′)	Fragment (bp)	RFLP Patterns	Restriction Enzymes	References
PCR-RFLP	Nos-16S-fw Nos-16S-rv	CGTAGACGCTATTCCCTAAGATT CTCCCAACTATACAGTACACCTCATA	488	*N. ceranae* 97, 118, 269 bp. *N. apis* 91, 131, 266 bp.	Pac I Nde I Msp I	[28]
PCR-RFLP	SSUres-f1	GCCTGACGTAGACGCTATTC	~400	*N. cerane* 104, 116, 177 bp. *N. apis* 91, 136, 175 bp.	Pac I Nde I Msp I	[29]
	SSUres-r1	GTATTACCGCGGCTGCTGG				

**Table 4 microorganisms-13-02501-t004:** Detection of *Nosema* spp. by real-time PCR.

Species	Primers’ Names	Sequence (5′ → 3′)	Fragment (bp)	Target	References
*N. apis*	NaFor	Fwd: CTAGTATATTTGAATATTT GTTTACAATGG	278	16S rRNA	[31]
*N. ceranae*	NcFor	Fwd: TATTGTAGAGAGGTGGGAGATT	316	16S rRNA	[31]
*Nosema* universal	UnivRev	Urev: GTC GCT ATG ATC GCT TGC C		16S rRNA	[31]
*N. apis*		Fwd: TGCAGATTTTGACGGAGATGA Rev: TGTACAATACCCATTATAGGACGA	138	RPB1	[34]
*N. ceranae*		Fwd: TCTTGTTCCTCCACCATCAGT Rev: TGTGTCAAATCATCTTCTGCTCT	75	RPB1	[34]
*N. bombi*		Fwd: GGAGAAATCTGTGAAAGTGGGT Rev: GGCTACTAGTCCCATTCCTTCT	81	RPB1	[34]
*N. ceranae*		Fwd: GGGATTACAAGTGCTTAGAGTGATT Rev: TGTCAAGCCCATAAGCAAGTG	65	Hsp70	[35]
*N. ceranae*	DQ486027 F DQ486027 R	Fwd: GGTTGGGAGAAGCCGTTACC Rev: ACCTGATCCAACGCAAATGCTA	103	16S rRNA	[36]
*N. apis*	U97150 F U97150 R	Fwd: GGAACCACCTTTTCTCCTACAAGCAA Rev: CCAAAAACTCCCAAGGAAAAACAAAAC	92	16S rRNA	[36]

**Table 5 microorganisms-13-02501-t005:** Overview of PCR conditions for the detection of *Nosema* species in different studies.

References	Type of PCR	Annealing Temperature	Duration	Number of Cycles	Total Mix Volume
[22]	duplex PCR	61.8 °C	30 s	30	50 μL
[24]	multiplex PCR	55 °C	30 s	35	10 μL
[39]	duplex PCR	58 °C	30 s	35	20 μL
[28]	RFLP PCR	53 °C	1 min	45	25 μL
[29]	RFLP PCR	48 °C	1 min	45	25 μL
[23]	multiplex PCR	61.8 °C	30 s	30	15 μL
[28]	duplex PCR	61.8 °C	30 s	30	25 μL
[27]	*N. apis* specific PCR	62 °C	1 min	40	20 μL
[38]	duplex PCR	62 °C	30 s	35	25 μL
[37]	multiplex PCR	46 °C	1 min	35	25 μL
[25]	multiplex PCR	55 °C	30 s	35	10 μL
[26]	multiplex PCR	56 °C	45 s	30	15 μL
[40]	duplex PCR	58 °C	45 s	35	25 μL

**Table 6 microorganisms-13-02501-t006:** Overview of thermal conditions of real-time PCR protocols for the detection of *Nosema* spp.

Step	Temperature (°C)	Time	Number of Cycles	Total Mix Volume
Forsgren and Fries (2010) [31] 20 µL				
Initial activation step	98	15 min	1	20 µL
PCR cycling			40	
Denaturation	98	5 s		
Annealing and elongation	63	10 s		
Analysis of the melting curve	65–95	10 s/step	-	
Babin et al. (2022) [34] 20 µL				
Initial activation step	95	3 min	1	20 µL
PCR cycling			45	
Denaturation	95	10 s		
Annealing of primers	60	30 s		
Elongation	72	25 s		
Termination of reaction	40	10 s	1	
Cilia et al. (2018) [35] 20 µL				
Initial activation step	95	10 min	1	20 µL
PCR cycling			40	
Denaturation	95	15 s		
Annealing of primers	56	60 s		
Traver a Fell (2011) [36] 20 µL				
Initialization step	50	2 min	1	20 µL
Initial activation step	95	10 min	1	
PCR cycling			40	
Denaturation	95	15 s		
Annealing/elongation	60	1 min		

**Table 7 microorganisms-13-02501-t007:** Components of the PCR mix and their amounts.

Components of the PCR Mix	Concentrations	Quantity
PCR water		11.5 µL
Firepol Master Mix (Solis Biodine)	1.5 mM MgCl_2_	4 µL
218MITOC-FOR/REV (specific for *N. ceranae*)	10 pmol	1 µL
321APIS-FOR/REV (specific for *N. apis*) of Martín-Hernández et al. (2007) [22]	10 pmol	1 µL
Template		2.5 µL

**Table 8 microorganisms-13-02501-t008:** Composition of the reaction mixture for real-time PCR detection of *Nosema* spp.

FastStart Universal SYBR Green Master (Roche):	12.5 μL
Specific primers (0.3 μM)	
APIS FOR (5′-GGGGCCATGTGTTTGACGTACTATGTA-3′)	0.5 μL
APIS REV (5′-GGGGGGCGTTTAAAAATGTGAACAACTATG-3′)	0.5 μL
PCR water	4.5 μL
Template	7 μL
FastStart Universal SYBR Green Master (Roche):	12.5 μL
Specific primers (0.3 μM)	
MITOC FOR (5′CGGCGACGATGATGATGATGAAAAATATTAA-3′)	0.5 μL
MITOC REV (5′-CCCGGTCATTCTCAAAAAAACCG-3’)	0.5 μL
PCR water	4.5 μL
Template	7 μL

**Table 9 microorganisms-13-02501-t009:** Program used for real-time PCR.

Step	Temperature (°C)	Time	Number of Cycles
Incubation	50 °C	2 min	1
Initial denaturation	95 °C	10 min	1
Denaturation	95 °C	15 s	
Hybridization	60 °C	1 min	
Melting	95 °C	15 s	1

**Table 10 microorganisms-13-02501-t010:** Comparison of the yield of individual methodologies by our workplace.

Detection Method	Total Number of Samples	Positive	Negative
Microscopy	500	115	385
Duplex PCR	500	107	393
Real-time PCR	500	110	390

## Data Availability

No new data were created or analyzed in this study. Data sharing is not applicable to this article.

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
