# Peer review of "An Overview of the Most Commonly Used Methods for the Detection of Nosema spp. in Honeybees"

_microorganisms, 2025, doi:10.3390/microorganisms13112501_

Round 1
Reviewer 1 Report
Comments and Suggestions for Authors
This is a REVIEW ARTICLE, not an Original (Research) article. The authors chose the wrong Article type, so the template inserted subtitles "2. Materials and Methods" (Line 51) and „3. Results“ (Line 146) that should be removed. I am unsure how they can change the article type at this stage, so I am asking the editors to assist the authors in changing the article type and get an adequate template for the Review article.
The aim of this manuscript is well written in Abstract (Line 24), but it should be given also at the end of the Introduction section.
The manuscript contains many errors, but I have tried to explain in detail how to correct them - see below.
Lines 18-19: I suggest to replace „bee faecal spores or the digestive tract by light microscopy“ with „bee digestive tract (macerated) by light microscopy“.
Lines 19-20: TEM and SEM have never replaced light microscopy. I suggest writing a separate sentence related to TEM and SEM, as follows: „Transmission electron microscopy (TEM) and scanning electron microscopy (SEM) are expensive methods that require skilled personnel, and were used only when high resolution was necessary“.
Lines 30-32: This sentence is wrong: „It functions as a superorganism that forms numerous communities, participating in pollination and the formation of diverse plant cultures.“
I suggest replacing it withthis one: „They live in colonies (consisting of several tens of thousands of individuals) that function as superorganisms. Their most important role is pollination of flowering plants, including fruit trees and crops, and therefore plays a major and indispensable role in agriculture as well as in the conservation of ecological biodiversity in general [1]“.
Line 34: It looks like you skipped the word "one"; in fact, instead „only the bee product“, you should write „only one bee product“.
Lines 37-40: Delete these two sentences, as I incorporated them above, at the beginning of the Introducton (because the pollination role is much more important than collecting bee products, so the importance of honey bees as pollinators should be emphasized before beekeeping benefits.
Lines 44-45: The sentence „Globally, N. apis and N. ceranae are currently the most prevalent pathogens infecting A. mellifera colonies of the genus Nosema“ should be replaced with „Globally, microsporidian species Nosema apis and N. ceranae are currently the most prevalent pathogenic microorganisms of A. mellifera [5–11].
Lines 45-46: „The two species are very similar at first sight, differing by slight differences in spore dimensions“ should be replaced with: „The spores of these two species are very similar at first sight; the differences between them could be distinguished only under Transmission Electron Microscopy - TEM in dimensions and number of polar filament coils (Fries et al., 1996), or by surface structure under the Scanning Electron Microscopy-SEM (Ptaszyńska et al., 2014).
- Fries I, Feng F, da Silva A, Slemenda SB, Pieniazek NJ (1996) Nosema ceranae sp. (Microspora, Nosematidae), morphological and molecular characterization of a microsporidian parasite of the Asian honey bee Apis cerana (Hymenoptera, Apidae), Eur. J. Protistol. 32(3), 356–365.
- Ptaszyńska AA, Borsuk G, Mułenko W, Demetraki-Paleolog J (2014) Differentiation of Nosema apis and Nosema ceranae spores under Scanning Electron Microscopy (SEM). J. Apicult. Res. 53(5), 537-544.
Line 51: Delete the subtitle "2. Materials and Methods" as it is not adequate for a review article. It is the title of the subchapter in the original papers where you describe the methodology you used. The subtitle „Methodologies used for the detection of Nosema spp.“ is OK, but I would expand it to read „Overview of methodologies used for the detection of Nosema spp.“.
Line 65: Please avoid the term "Recent", since Fries et al. (1996) used TEM for detailed analysis of N. apis and N. ceranae spores almost 30 years ago.
Lines 68-71: Please be careful. In ref. 20 the subject is Nosema bombycis. Differentiation between N. apis and N. ceranae by TEM was first done and published by Fries et al. (1996) Eur. J. Protistol, 32 (3), 356–365.
Line 93 and Line 97: You wrote „The tables“. This is incorrect; you should refer to the specific table.
Line 94, instead of „multiplex PCR, standard and RFLP PCR methods“ you should write „multiplex (including duplex) PCR, single species-specific PCR and RFLP PCR methods).
In the title of Table 2, within Table 2 and within Table 5: you wrote "standard" PCR, whereby by that name in Table 2 and Table 5, it is obvious that you mean N. apis specific PCR. Therefore, you should replace the word „standard“ with „N. apis specific PCR“ in the title of Table 2, within Table 2 and within Table 5:
Table 1. must be significantly corrected and shortened. The reasons are:
- These primers for duplex PCR:
218MITOC-FOR 5_-CGGCGACGATGTGATATGAAAATATTAA-3;
218MITOC-REV 5_-CCCGGTCATTCTCAAACAAAAAACCG-3 (218–219 bp, specific for N. ceranae) and
321APIS-FOR 5_-GGGGGCATGTCTTTGACGTACTATGTA-3;
321APIS-REV 5_-GGGGGGCGTTTAAAATGTGAAACAACTATG-3 (321 bp, specific for N. apis)
were originally designed and published by Martín-Hernández et al. (2007), ref. No. 21 in your manuscript and originally named: 218MITOC-FOR, 218MITOC-REV (please correct the names in the first rows of your Table).
Thus, you should not attribute these primers to other authors (ref. Nos. 22, 25, 26, 27 and 30) because all of them only used primers from Martín-Hernández et al. (2007), who have the intellectual property rights to these primers. Each of these references (refs. Nos. 22, 25, 26, 27 and 30) clearly states that these primers were designed by Martín-Hernández et al. (2007). Therefore, you should not write the same primers 6 times in Table 1. After all, there are a huge number of papers in which these primers have been used, so it is impossible and unnecessary to list them all.
- You made a mistake: Fragment (bp) in ref. No. 25 is not 488. As I already said, in ref. No. 25, for duplex PCR, primers from Martín-Hernández et al. (2007) were used, and a 488 bp fragment was obtained using Nosema genus-specific primers in the first step of the PCR-RFLP method.
- You should add the third pair of primers (or more) where you stated that it was a multiplex PCR.
- Since the target sequence was always located within 16S rRNA, delete the columnn „Target“ and incorporate that fact into the Table 1 title, eg. „Detection of Nosema by multiplex PCR of partial 16S rRNA gene“.
Table 3: Incorrect title. I suggest this one: „Differentiation between Nosema species based on PCR–RFLPs of partial 16S rRNA gene“.
Table 3: Delete the column „Species“ and the column „Target“
Table 3. You wrote the wrong fragment length (222 bp), because primer pair SSU-res-f1/SSU-res-r1 enables amplification of the fragment of ~400 bp.
Table 3. There is one more PCR-RFLP method with another pair of primers (Nos-16S-fw/Nos-16S-rv) designed and published by Stevanovic et al. (2011) – the ref. No. 25 in your manuscript. The primers are Nosema genus-specific and their sequences are: Nos-16S-fw: 5'-CGTAGACGCTATTCCCTAAGATT-3'/Nos-16S-rv: 5'-CTCCCAACTATACAGTACACCTCATA-3', and the fragment they amplify is 488 bp long. Therefore, please insert a new row in Table 3, with the details related to Nos-16S-fw/Nos-16S-rv PCR-RFLP.
Finally, the names of restriction enzymes and RFLP patterns (for both PCR-RFLP methods) should be added to Table 3.
Table 4: Add the colony with the primers’ names.
Table 4: „Fwd“ and „Rev“ are missing at the first pair of primers.
Table 5: Write „Annealing temperature“ instead of „Temperature“.
Line 146. Delete the subtitle "3. Results" as it is not adequate for a review article. The content of this subsection contains your materials, methods and the results, so it is OK just to leave the subtitle: „Comparison of methodologies by our department“.
Table 7: Primers’ sequences are not necessary. Please write that you used the primers 218MITOC-FOR/REV (specific for N. ceranae) and 321APIS-FOR/REV (specific for N. apis) of Martín-Hernández et al. (2007). Another important correction I suggest: write FINAL CONCENTRATIONS (instead of working volumes) of each component.
Table 8: RT PCR denotes Reverse Transcription-PCR, so you should replace RT PCR with real-time PCR.
Lines 153-174: I did not find a word related to real-time PCR you performed, and you gave the results in Table 8. Did you forget to describe the real-time PCR performed by your department?
Author Response
This is a REVIEW ARTICLE, not an Original (Research) article. The authors chose the wrong Article type, so the template inserted subtitles "2. Materials and Methods" (Line 51) and „3. Results“ (Line 146) that should be removed. I am unsure how they can change the article type at this stage, so I am asking the editors to assist the authors in changing the article type and get an adequate template for the Review article.
The aim of this manuscript is well written in Abstract (Line 24), but it should be given also at the end of the Introduction section.
The manuscript contains many errors, but I have tried to explain in detail how to correct them - see below.
Lines 18-19: I suggest to replace „bee faecal spores or the digestive tract by light microscopy“ with „bee digestive tract (macerated) by light microscopy“.
Lines 19-20: TEM and SEM have never replaced light microscopy. I suggest writing a separate sentence related to TEM and SEM, as follows: „Transmission electron microscopy (TEM) and scanning electron microscopy (SEM) are expensive methods that require skilled personnel, and were used only when high resolution was necessary“.
Lines 30-32: This sentence is wrong: „It functions as a superorganism that forms numerous communities, participating in pollination and the formation of diverse plant cultures.“
I suggest replacing it withthis one: „They live in colonies (consisting of several tens of thousands of individuals) that function as superorganisms. Their most important role is pollination of flowering plants, including fruit trees and crops, and therefore plays a major and indispensable role in agriculture as well as in the conservation of ecological biodiversity in general [1]“.
Line 34: It looks like you skipped the word "one"; in fact, instead „only the bee product“, you should write „only one bee product“.
Lines 37-40: Delete these two sentences, as I incorporated them above, at the beginning of the Introducton (because the pollination role is much more important than collecting bee products, so the importance of honey bees as pollinators should be emphasized before beekeeping benefits.
Lines 44-45: The sentence „Globally, N. apis and N. ceranae are currently the most prevalent pathogens infecting A. mellifera colonies of the genus Nosema“ should be replaced with „Globally, microsporidian species Nosema apis and N. ceranae are currently the most prevalent pathogenic microorganisms of A. mellifera [5–11].
Lines 45-46: „The two species are very similar at first sight, differing by slight differences in spore dimensions“ should be replaced with: „The spores of these two species are very similar at first sight; the differences between them could be distinguished only under Transmission Electron Microscopy - TEM in dimensions and number of polar filament coils (Fries et al., 1996), or by surface structure under the Scanning Electron Microscopy-SEM (Ptaszyńska et al., 2014).
- Fries I, Feng F, da Silva A, Slemenda SB, Pieniazek NJ (1996) Nosema ceranae sp. (Microspora, Nosematidae), morphological and molecular characterization of a microsporidian parasite of the Asian honey bee Apis cerana (Hymenoptera, Apidae), Eur. J. Protistol. 32(3), 356–365.
- Ptaszyńska AA, Borsuk G, Mułenko W, Demetraki-Paleolog J (2014) Differentiation of Nosema apis and Nosema ceranae spores under Scanning Electron Microscopy (SEM). J. Apicult. Res. 53(5), 537-544.
Line 51: Delete the subtitle "2. Materials and Methods" as it is not adequate for a review article. It is the title of the subchapter in the original papers where you describe the methodology you used. The subtitle „Methodologies used for the detection of Nosema spp.“ is OK, but I would expand it to read „Overview of methodologies used for the detection of Nosema spp.“.
Line 65: Please avoid the term "Recent", since Fries et al. (1996) used TEM for detailed analysis of N. apis and N. ceranae spores almost 30 years ago.
Lines 68-71: Please be careful. In ref. 20 the subject is Nosema bombycis. Differentiation between N. apis and N. ceranae by TEM was first done and published by Fries et al. (1996) Eur. J. Protistol, 32 (3), 356–365.
Line 93 and Line 97: You wrote „The tables“. This is incorrect; you should refer to the specific table.
Line 94, instead of „multiplex PCR, standard and RFLP PCR methods“ you should write „multiplex (including duplex) PCR, single species-specific PCR and RFLP PCR methods).
In the title of Table 2, within Table 2 and within Table 5: you wrote "standard" PCR, whereby by that name in Table 2 and Table 5, it is obvious that you mean N. apis specific PCR. Therefore, you should replace the word „standard“ with „N. apis specific PCR“ in the title of Table 2, within Table 2 and within Table 5:
Table 1. must be significantly corrected and shortened. The reasons are:
- These primers for duplex PCR:
218MITOC-FOR 5_-CGGCGACGATGTGATATGAAAATATTAA-3;
218MITOC-REV 5_-CCCGGTCATTCTCAAACAAAAAACCG-3 (218–219 bp, specific for N. ceranae) and
321APIS-FOR 5_-GGGGGCATGTCTTTGACGTACTATGTA-3;
321APIS-REV 5_-GGGGGGCGTTTAAAATGTGAAACAACTATG-3 (321 bp, specific for N. apis)
were originally designed and published by Martín-Hernández et al. (2007), ref. No. 21 in your manuscript and originally named: 218MITOC-FOR, 218MITOC-REV (please correct the names in the first rows of your Table).
Thus, you should not attribute these primers to other authors (ref. Nos. 22, 25, 26, 27 and 30) because all of them only used primers from Martín-Hernández et al. (2007), who have the intellectual property rights to these primers. Each of these references (refs. Nos. 22, 25, 26, 27 and 30) clearly states that these primers were designed by Martín-Hernández et al. (2007). Therefore, you should not write the same primers 6 times in Table 1. After all, there are a huge number of papers in which these primers have been used, so it is impossible and unnecessary to list them all.
- You made a mistake: Fragment (bp) in ref. No. 25 is not 488. As I already said, in ref. No. 25, for duplex PCR, primers from Martín-Hernández et al. (2007) were used, and a 488 bp fragment was obtained using Nosema genus-specific primers in the first step of the PCR-RFLP method.
- You should add the third pair of primers (or more) where you stated that it was a multiplex PCR.
- Since the target sequence was always located within 16S rRNA, delete the columnn „Target“ and incorporate that fact into the Table 1 title, eg. „Detection of Nosema by multiplex PCR of partial 16S rRNA gene“.
Table 3: Incorrect title. I suggest this one: „Differentiation between Nosema species based on PCR–RFLPs of partial 16S rRNA gene“.
Table 3: Delete the column „Species“ and the column „Target“
Table 3. You wrote the wrong fragment length (222 bp), because primer pair SSU-res-f1/SSU-res-r1 enables amplification of the fragment of ~400 bp.
Table 3. There is one more PCR-RFLP method with another pair of primers (Nos-16S-fw/Nos-16S-rv) designed and published by Stevanovic et al. (2011) – the ref. No. 25 in your manuscript. The primers are Nosema genus-specific and their sequences are: Nos-16S-fw: 5'-CGTAGACGCTATTCCCTAAGATT-3'/Nos-16S-rv: 5'-CTCCCAACTATACAGTACACCTCATA-3', and the fragment they amplify is 488 bp long. Therefore, please insert a new row in Table 3, with the details related to Nos-16S-fw/Nos-16S-rv PCR-RFLP.
Finally, the names of restriction enzymes and RFLP patterns (for both PCR-RFLP methods) should be added to Table 3.
Table 4: Add the colony with the primers’ names.
Table 4: „Fwd“ and „Rev“ are missing at the first pair of primers.
Table 5: Write „Annealing temperature“ instead of „Temperature“.
Line 146. Delete the subtitle "3. Results" as it is not adequate for a review article. The content of this subsection contains your materials, methods and the results, so it is OK just to leave the subtitle: „Comparison of methodologies by our department“.
Table 7: Primers’ sequences are not necessary. Please write that you used the primers 218MITOC-FOR/REV (specific for N. ceranae) and 321APIS-FOR/REV (specific for N. apis) of Martín-Hernández et al. (2007). Another important correction I suggest: write FINAL CONCENTRATIONS (instead of working volumes) of each component.
Table 8: RT PCR denotes Reverse Transcription-PCR, so you should replace RT PCR with real-time PCR.
Lines 153-174: I did not find a word related to real-time PCR you performed, and you gave the results in Table 8. Did you forget to describe the real-time PCR performed by your department?
Dear Reviewer 1,
First of all, I would like to thank you on behalf of myself and the author team. We are glad that you took the time to read our manuscript and gave us valuable advice and insights on how to improve it.
Based on your recommendations, we are addressing the points and their corrections below:
Point 1: wrong Article type - we are communicating with the assigned editor
Point 2: aiml of the manuscript - we also stated it at the end of the introduction
Point 3: Lines 18-19 - corrected based on your suggestion
Point 4: Lines 19-20 - corrected based on your suggestion
Point 5: Lines 30-32 - corrected based on your suggestion
Points 6 and 7: Lines 34, 37-40 - corrected based on your suggestion
Point 8: Lines 44-45 - corrected based on your suggestion
Point 9: Lines 45-46 - corrected based on your suggestion
Point 10: Line 65 - the term "Recent" replaced by the term "Previous"
Point 11: Lines 68-71 - since the subject was Nosema bombycis, the paragraph was removed as an incorrect data in the comparison of N. apis and N. ceranae
Point 12: Line 93 and line 97 - supplemented with numbers of tables to which we refer
Point 13: Line 94 - corrected based on your suggestion
Point 14: Tables 2 and 5 - corrected based on your suggestion
Point 15: Table 1 - completely corrected and supplemented based on your recommendations
Point 16: Table 3 - corrected and supplemented based on your recommendations
Point 17: Table 4 - modified and supplemented, despite this, we were unable to find all the names for the listed primers
Point 18: Table 5 - "Temperature" corrected to "Annealing temperature".
Point 19: Table 7 - corrected based on your suggestion, volumes replaced with concentrations
Point 20: Table 8 - now Table 10, RT PCR replaced with real-time PCR
Point 21: Lines 153-174 - text supplemented with the performed real-time PCR and also with Tables to make the text clearer
Best regards, the author team
Reviewer 2 Report
Comments and Suggestions for Authors
Although the manuscript has a scientific value and most of included information are presented in a logical way it needs some corrections.
Authors should explicitly describe their article search methodology, including the search strategy employed, the rationale behind their selection criteria, and the specific databases consulted. A transparent account of the search process enables readers to assess the comprehensiveness and reproducibility of the literature review. The authors should specify which academic databases were searched (e.g., PubMed, Web of Science, Scopus, IEEE Xplore). Furthermore, the authors should justify their inclusion and exclusion criteria, explaining the decision-making process that guided article selection. This methodological transparency is essential for establishing the credibility and rigor of the research, as it allows other researchers to replicate the search procedure and verify the findings. The rationale for choosing particular databases and search parameters should be grounded in the specific research objectives and the nature of the subject matter under investigation.
Additionally, authors should incorporate emerging diagnostic methodologies into their review, such as the novel loop-mediated isothermal amplification (LAMP) assays presented in recent publications, including the work on rapid detection and differentiation of Nosema apis and N. ceranae in honeybees (Loop-mediated isothermal amplification (LAMP) assays for rapid detection and differentiation of Nosema apis and N. ceranae in honeybees
(https://onlinelibrary.wiley.com/doi/10.1111/1574-6968.12521?msockid=1d8e42f7e079655d2f0d56a0e1a064a7). The inclusion of innovative detection techniques demonstrates the authors' awareness of current advances in the field and ensures that the literature review encompasses both established and cutting-edge approaches to pathogen identification and diagnostic testing.
Author Response
Although the manuscript has a scientific value and most of included information are presented in a logical way it needs some corrections.
Authors should explicitly describe their article search methodology, including the search strategy employed, the rationale behind their selection criteria, and the specific databases consulted. A transparent account of the search process enables readers to assess the comprehensiveness and reproducibility of the literature review. The authors should specify which academic databases were searched (e.g., PubMed, Web of Science, Scopus, IEEE Xplore). Furthermore, the authors should justify their inclusion and exclusion criteria, explaining the decision-making process that guided article selection. This methodological transparency is essential for establishing the credibility and rigor of the research, as it allows other researchers to replicate the search procedure and verify the findings. The rationale for choosing particular databases and search parameters should be grounded in the specific research objectives and the nature of the subject matter under investigation.
Additionally, authors should incorporate emerging diagnostic methodologies into their review, such as the novel loop-mediated isothermal amplification (LAMP) assays presented in recent publications, including the work on rapid detection and differentiation of Nosema apis and N. ceranae in honeybees (Loop-mediated isothermal amplification (LAMP) assays for rapid detection and differentiation of Nosema apis and N. ceranae in honeybees
(https://onlinelibrary.wiley.com/doi/10.1111/1574-6968.12521?msockid=1d8e42f7e079655d2f0d56a0e1a064a7). The inclusion of innovative detection techniques demonstrates the authors' awareness of current advances in the field and ensures that the literature review encompasses both established and cutting-edge approaches to pathogen identification and diagnostic testing.
Dear Reviewer 2,
First of all, I would like to thank you on behalf of myself and the author team. We are glad that you took the time to read our manuscript and gave us valuable advice and insights on how to improve it.
Based on your recommendations, we address the points and their modifications below:
We have also included the loop-mediated isothermal amplification (LAMP) methodology in our review as a diagnostic methodology that you recommended to us.
Best regards, the author team

Round 2
Reviewer 1 Report
Comments and Suggestions for Authors
The authors significantly and properly corrected the manuscript. I found only some errors that I had missed last time.
In Table 1, one mistake remained: The third pair of primers is the same as the first pair. In fact, Ostroverkhova et al. (2020). used the primers of Martín-Hernández et al. (2007). Please check the study of Ostroverkhova et al. (2020) and you will see that they referred to Martín-Hernández et al. (2007) for primers. Also, the names of the primers are identical (321APIS-FOR, 321APIS-REV and 218MITOC-FOR, 218MITOC-REV) as those of Martín-Hernández et al. (2007) primers, as well as the length of target fragments (321 bp for Nosema apis and 218 bp for Nosema ceranae), although you mistakenly wrote 128 bp (instead of 218) in the column „Fragment (bp)“.
Thus, third pair of primers should be removed from Table 1.
Line 108: „quantitative real-time PCR (qRT-PCR)“. This is wrong. RT-PCR denotes Reverse Transcription-PCR, and real-time PCR should not be abbreviated as RT-PCR. Another error is writing quantitative real-time, because quantitative has to be real-time, so you only need to write one: either quantitative PCR (qPCR) or real-time PCR (without abbreviation RT, because RT PCR denotes Reverse Transcription).
Line 111, Line 138, Line 141, the title of Table 4, the title of Table 6: Again, „qRT-PCR“ – please, see the previous comment and correct (write „real-time PCR“).
When it comes to the research design, I want to explain why I chose the option „Must be improved“. I remind editors that the type of the article should be changed to „Review“ and then the design will be appropriate.
Author Response
In Table 1, one mistake remained: The third pair of primers is the same as the first pair. In fact, Ostroverkhova et al. (2020). used the primers of Martín-Hernández et al. (2007). Please check the study of Ostroverkhova et al. (2020) and you will see that they referred to Martín-Hernández et al. (2007) for primers. Also, the names of the primers are identical (321APIS-FOR, 321APIS-REV and 218MITOC-FOR, 218MITOC-REV) as those of Martín-Hernández et al. (2007) primers, as well as the length of target fragments (321 bp for Nosema apis and 218 bp for Nosema ceranae), although you mistakenly wrote 128 bp (instead of 218) in the column „Fragment (bp)“.
Thus, third pair of primers should be removed from Table 1.
Line 108: „quantitative real-time PCR (qRT-PCR)“. This is wrong. RT-PCR denotes Reverse Transcription-PCR, and real-time PCR should not be abbreviated as RT-PCR. Another error is writing quantitative real-time, because quantitative has to be real-time, so you only need to write one: either quantitative PCR (qPCR) or real-time PCR (without abbreviation RT, because RT PCR denotes Reverse Transcription).
Line 111, Line 138, Line 141, the title of Table 4, the title of Table 6: Again, „qRT-PCR“ – please, see the previous comment and correct (write „real-time PCR“).
When it comes to the research design, I want to explain why I chose the option „Must be improved“. I remind editors that the type of the article should be changed to „Review“ and then the design will be appropriate.
Dear Reviewer 1,
First of all, I would like to thank you on behalf of myself and the author team. We are glad that you took the time to read our manuscript again and provided us with valuable advice and insights on how to improve it.
Based on your recommendations, we are addressing the following points and their corrections:
Point 1: In Table 1 we removed the duplicate study
Point 2: line 108- we edited the title
Point 3: Line 111, Line 138, Line 141, the title of Table 4, the title of Table 6 - also edited
Point 4: "Review" this status is being resolved with the editor
Best regards, the author team

Reviewer 2 Report
Comments and Suggestions for Authors
The authors have made appropriate corrections and changes based on the reviewers' comments, and the manuscript is acceptable for publication.
Author Response
The authors have made appropriate corrections and changes based on the reviewers' comments, and the manuscript is acceptable for publication.
Dear Reviewer 2
We are very grateful for your positive opinion of our manuscript. We would like to thank you for your evaluation and recommendation. We are glad that you are satisfied with our editing.
Best regards, the author team
